# Deep Deformation Based on Feature-Constraint for 3D Human Mesh Correspondence

## Abstract

In this study, we address the challenges in mesh correspondence for various types of complete or single-view human body data. The parametric human model has been widely used in various human-related applications and in 3D human mesh correspondence because it provides sufficient scope to modify the resulting model. In contrast to prior methods that optimize both the correspondences and human model parameters (pose and shape), some of the recent methods directly deform each vertex of a parametric template by processing the point clouds that represent the input shapes. This allows the models to have more accurate representations of the details while maintaining the correspondence. However, we identified two limitations in these methods. First, it is difficult for the transformed template to completely restore the input shapes using only a pointwise reconstruction loss. Second, they cannot deform the template to a single-view human body from the depth camera observations or infer the correspondences between various forms of input human bodies. In representation learning, one of the main challenges is to design appropriate loss functions for supervising features with different abilities. To address this, we introduce the feature constraint deformation network (FCD-Net), which is an end-to-end deep learning approach that identifies 3D human mesh correspondences by learning various shape transformations from a predetermined template. The FCD-Net is implemented by an encoder–decoder architecture. A global feature encoded from the input shape and a decoder are used to deform the template based on the encoded global feature. We simultaneously input the complete shape and single-view shape into the encoder and closely constrain the features to enable the encoder to learn more robust features. Meanwhile, the decoder generates a completely transformed template with higher promise by using the complete shape as the ground truth, even if the input is single-view human body data. We conduct extensive experiments to validate the effectiveness of the proposed FCD-Net on four types of single-view human body data, both from qualitative and quantitative aspects. We also demonstrate that our approach improves the state-of-the-art results on the difficult "FAUST-inter" and "SHREC'19" challenges, with average correspondence errors of 2.54 cm and 6.62 cm, respectively . In addition, the proposed FCD-Net performs well on real and unclean point clouds from a depth camera.

## 1 Introduction

The rapid development of 3D sensor devices has led to tremendous growth in the field of 3D vision technologies. An essential application of 3D vision technology is the 3D shape correspondence and deformation(Huang & Fang (2021)), which attempts to establish reliable correspondences between two 3D shapes(Klatzow et al. (2022); Sahillioglu & Yemez (2010); Huang et al. (2017)) and is a hot research topic in 3D vision. In contrast to registrations of scenes or objects (Gojcic et al. (2020); Segal et al. (2009); Ao et al. (2021)) that only involves rigid deformations, such as rotations and translations, estimating the correspondences on articulated human bodies requires flexible, complex, non-rigid deformations and pose variations(Serafin & Grisetti (2015); Bhatnagar et al. (2020); Groueix et al. (2018)). This makes the correspondence process more challenging.

Model-driven shape reconstruction and matching methods for articulated humans utilize a parametric body template model (e.g., a skinned multiperson linear (SMPL ) (Loper et al. (2015)) or shape completion and animation for people (SCAPE) (Anguelov et al. (2005)) model) as a geometrical prior, and optimize or learn the parameters to deform the template, typically its poses and shapes. The deformed models have the same vertex definition, definite semantic information, and same face connections as the template. This makes the correspondence problem easier than when using methods that require finding associations between a variable and large number of points through an optimization strategy to minimize an objective function. However, the low dimensional parameters (shape and pose) limit the description of the details. Researchers have proposed SMPL model + displacement approaches to increase the details of the model (Bhatnagar et al. (2020)). Other works have combined a parametric model with the deep implicit function to realize free-form human body reconstruction at arbitrary resolutions (Huang et al. (2020); He et al. (2021)). However , SMPL + displacement methods require two stages to output the final results, making them prone to error at two levels. Free-form implicit functions can lose the semantic information and correspondences of the points. Some recent works Wang et al. (2019) have directly deformed each vertex of a parametric template based on a global feature coded from the input shape. This allows the models to provide a more accurate representation of details while maintaining the correspondence of the parametric model. However, we identified two limitations in these methods. First, it is difficult for the transformed template to completely restore the input shapes using only a pointwise reconstruction loss. Second, they can only deform the template to a complete human body as obtained by scanning (or via reconstruction by other methods(Choe et al. (2021))). In this study , we focus on deforming the template to single-view 3D human shapes from depth camera images and inferring the correspondences between various forms of the human body. A single-view shape is the most easily obtained 3D data form, owing to the development of low cost, low power consumption , and low-price RGBD cameras. In Groueix et al. (2018), a point-based neural network was used to learn a global feature from an input shape. The global feature stores all of the information of the input shape for directing the template deformation. In our view, the global feature should not only accurately represent the complete human shape but should also facilitate recovering the invisible part(s) when the input is a single-view shape. In representation learning, one of the main challenges is to design appropriate loss functions for supervising features with different abilities (discriminative, expressive, or restorative). Motivated by Deng et al. (2019), we attempted to penalize the loss in the feature space, and specifically, to penalize the angular difference between the deep features obtained from the single-view shape and complete human shape. This can be achieved by simultaneously inputting the complete and single-view data of one object during the training process.

Therefore, we propose a framework that is suitable for searching for the correspondence relationships between various types of complete or single-view human bodies by processing the point clouds that represent the input shapes. We call this framework the feature constraint deformation network (FCD-Net). The FCD-Net is designed with an encoder-decoder structure. The encoder comprises a deep neural network for generating global features representing the input shapes. Then, a shape deformation network learns to deform a template as guided by the encoded global feature to align with the target shape as described in detail in Section .3 .

We train our FCD-Net with single-view and complete shapes as input shape pairs, as supervised by the known correspondences between the input shapes and template. These are generally explicit when they are both generated by the same parametric model. During testing, only one type of observation is needed, i.e., single-view or complete. The correspondences between the various types of inputs can be realized under a unified framework. To demonstrate the advantages of the FCD-Net, first, we show that the FCD-Net can achieve single-view 3D human body correspondence. Then, we test FCD-Net on finding the correspondences for scanned 3D humans on several public datasets. The FCD-Net achieves state-of-the-art results in the "INTER" challenge of the FAUST dataset with an average correspondence error of 2.54 cm, and in the SHREC'19 challenge with an average correspondence error of 6.62 cm.

## 2 RELATED WORKS

The traditional search for correspondence between 3D human body models is often conducted through regression optimization methods or function mapping methods. The normal iterative closest point (NICP)(Serafin & Grisetti (2015))algorithm represents the corresponding relationship between a source model and target model by establishing a complex mathematical relationship be-

tween the source and target model point sets, and then finds the corresponding points through continuous iteration. In addition to constraining the distances between points, the NICP adds constraints on the normal vector and curvature of the surface where the point cloud is located for the model matching, thereby improving the accuracy of the alignment. Yao et al. used Welsch's function to optimize an objective function; this changed the nonconvex discontinuity of the objective function and enabled the use of the majorize-minimize algorithm with the limited-memory Broyden–Fletcher–Goldfarb–Shanno algorithm. This improved the efficiency and accuracy in solving the correspondence(Yao et al. (2020)). Optimization-based methods have strict mathematical logic to ensure their convergence and interpretability, but they often need good initialization. Therefore, features, such as the heat kernel signature (Sun et al. (2009)), ware kernel signature (Aubry et al. (2011)), and orientation histogram, are widely used in non-rigid 3D shape processing tasks, and each identified key point is represented by a feature vector (Tombari et al. (2010)). These features are often used as a priori knowledge as a way to obtain a more reliable correspondence. However, methods that require additional computation to obtain prior knowledge are sensitive to problems, such as priori knowledge, noise, and voids.

A function mapping method transforms the process of finding the corresponding relationship between the source model and target model into function mapping, and it considers the source model and target model as changes between matrices. By adding normal constraints, approximation function constraints, and other constraints, the quality of the corresponding relationship is improved (Ovsjanikov et al. (2016); Vestner et al. (2017)). However, most of these methods require a large amount of computation or have strict requirements for inputs (Nogneng et al. (2018); Rodolà et al. (2015)). Another common treatment comprises a combination of function mapping and spectral domain; this has achieved good results on synthetic data, but is less effective in practice and requires a longer inference time (Ginzburg & Raviv (2020); Roufosse et al. (2019)).

AlexNet learned 3D features from RGBD datasets (Zeng et al. (2017)), and Ppfnet(Deng et al. (2018)) used a local point pair feature based on a neighborhood point distribution and fed it into a network for deep feature learning. 3DSmoothNet (Gojcic et al. (2019)) proposed rotation-invariant handcrafted features and fed them into deep neural networks for feature learning. The 3DN (Wang et al. (2019)) network keeps the network topology of the source model unchanged through differentiable network sampling operators, and an excellent corresponding relationship can be obtained between source and target models of different densities. Donati et al. used a hybrid network structure to learn feature descriptors by using feature extractors in the spatial domain (Donati et al. (2020)). After feature extraction, they were sent to a spectral domain function mapping layer to find the correspondences. Wang et al. transferred local features from an image space to a UV space and established a new UV mapping to obtain the correspondences between the images and 3D models (Zeng et al. (2020)). Correnet3D (Zeng et al. (2021)) used a corresponding learning module to compute the optimal matching matrices for source and target models with different poses and modeled the matching problem as a minimized cost function solution for further optimization. The above methods have achieved excellent results in 3D human model deformation and the corresponding relationship searches. However, these methods are usually proposed for matching complete 3D human models; there are relatively few methods applicable to single-view human body data collected by a single RGB-D camera. Our proposed method can identify the deformations and corresponding relationships between a single-view point cloud and complete human body, and obtain state-of-the art results.

## 3 METHOD

In this study, we aim to identify the shape correspondence for a single-view human body. To this end, we propose the feature constraint deformation network , which takes single-view point clouds of the human body as input. Inspired by Groueix et al. (2018), our network uses the parameterized human body as an intermediate bridge for the correspondence search. Compared with 3D-CODED data, when learning the reconstruction of the human shape model, our method takes the single-view human body as input, and it uses the complete human body model as a ground truth to supervise the network training in the pointwise and feature spaces. The obtained network can better realize the deformation and reconstruction of the single-view human body. Moreover, as the shape encoder network has seen the complete human body shape of the ground truth, the single-view point cloud correspondence network can realize the reconstruction and deformation of the complete human body

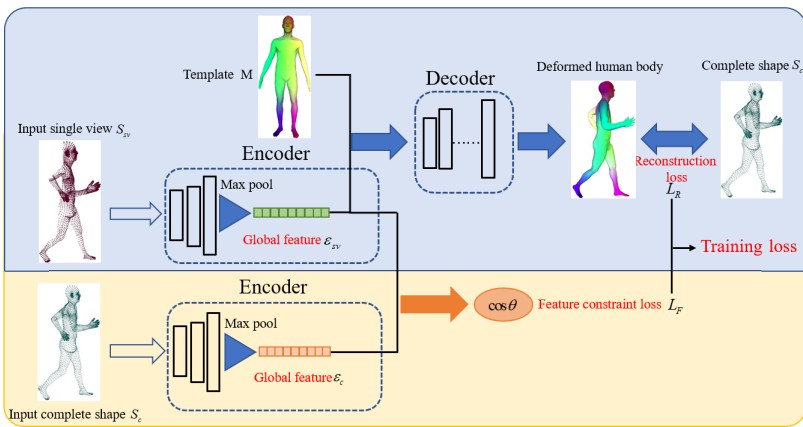

Figure 1: Overview of the learning procedure

model simultaneously. The learning and inference procedures of our method are shown in Fig. 1 and 3, and details are provided below.

### 3.1 LEARNING SINGLE-VIEW HUMAN BODY RECONSTRUCTION BY TEMPLATE DEFORMATION

It is a difficult and time-consuming task to directly obtain the correspondence relationship between two 3D human body shapes (source model and target model). The proposed method selects a parameterized human body model with a standard pose as a template, and then deforms it into source and target models with arbitrary poses and body shapes, respectively. The relationship between the corresponding points of the generated source model and target model is already established through the template, making the corresponding point detection relatively easy. To achieve this goal, we first design the shape deformation network. This network takes all of the vertices of the template as input and outputs the deformed vertex coordinates. To obtain guidance on the deformation, we must condition our observations (single-view human point clouds in our method) on the input. Therefore, the mapping relationship for shape deformation network is as follows:

$$D_\theta\left(p_j; x\right) \to q_j \tag{1}$$

where, $p_j$ denotes the $j^{\text{th}}$ vertex of the template, and $x$ denotes the observations.

The single-view human point cloud data in the original 3D space has great redundancy; therefore, we first adopt point cloud feature extraction $E_\phi$ to encode the single-view human point cloud, input all of the 3D point cloud coordinates, and output a one-dimensional feature vector with a predefined length. This feature is taken as the observed quantity in Equation (1) and in the human template as an input pair to output the deformed human shape vertices.

We achieve the above functions by training the coding-decoding network shown in the blue part of Fig. 1. The encoder adopts the simplified PointNet network to extract the features of the input point cloud. After a series of one-dimensional convolutions with scales of 64, 128, and 1024, features with 1024 dimensions are obtained through the maximum pooling layer. The decoder corresponds to the shape deformation network in Equation(1), and is implemented by a simple multilayer perceptron. The specific hidden layer sizes are 1024, 512, 254, and 128, i.e., the same as in Groueix et al. (2018). The 1024-dimensional features extracted by the encoder network are combined with the template body to complete the shape deformation. In the specific implementation, the human body template adopts the complete human body model, and the complete human body corresponding to the single-view point cloud is taken as the reconstructed ground truth. Thus, the proposed method can achieve deformation of the data of the visible view simultaneously . The missing data at the self-occluded regions is completed and the completed part is deformed. The reconstruction loss is defined as follows:

$$L_{\text{R}}(\phi, \theta) = \sum_{i=1}^{N}\sum_{j=1}^{P}\left|D_\theta\left(p_j; E_\phi\left(S_{sv}^{(i)}\right)\right) - q_j^{(i)}\right|^2 \tag{2}$$

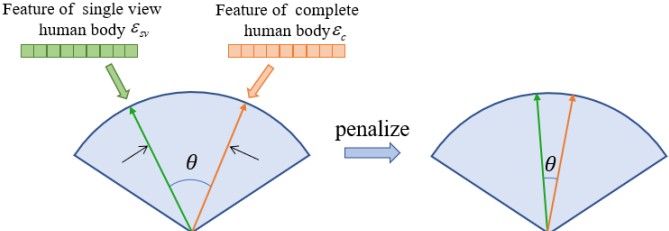

Figure 2: Feature constraints between complete and single view human body

Here $p_j \in M$ , where,$M$ represents the parameterized 3D human body model, $S_{sv}^{(i)}$ represents the $i^{th}$ single-view human shape in the input encoder network; $q_j^{(i)} \in S_c^{(i)}$ where$S_c^{(i)}$ represents the complete human body model corresponding to the input single-view human body model, $P$ represents the number of vertices, and $N$ represents the number of training samples.

The currently available methods for body shape correspondence (Zhou et al. (2020); Deprelle et al. (2019); Zeng et al. (2020)) can only achieve a correspondence between two complete bodies. The proposed method enables a determination of the correspondence between two single-view body shapes. Moreover, in addition to the need to estimate the shape correspondence between two single-view body shapes, there are many applications that require estimations of the shape correspondence between a single-view body shape and full-body shape. If our network only uses single-view body shapes as observations, it is will be unable to appropriately handle the deformation and reconstruction of a complete human body. Therefore, in addition to generating the deformed intact human body from the single-view human shape by supervising the network in the reconstruction space on a point-by-point basis, we also propose a feature-space constraint strategy (shown in yellow in Fig. 1). In this strategy, we input the intact human shape as a ground truth into the same shape encoder network, and then constrain the single-view human shape and intact human shape in the feature space as close as possible, so as to use the same network framework for both the single-view and full-body shapes. In addition, the proposed feature constraint strategy can also better guide the complementation and deformation of the single-view human shape, as shown in Fig. 2. The feature similarity is measured using the cosine distance, and the specific feature constraint loss is calculated as follows:

$$L_F(\phi) = \sum_{i=1}^{N} \left( 1 - \frac{\varepsilon_{sv}^{(i)} \cdot \varepsilon_c^{(i)}}{\left\| \varepsilon_{sv}^{(i)} \right\| \left\| \varepsilon_c^{(i)} \right\|} \right) \tag{3}$$

The total loss function used for optimizing the parameters of FCD is constructed as follows:

$$\text{Loss}(\phi, \theta) = (1 - b)L_R + bL_F \tag{4}$$

Here, $b$ is a scale coefficient for controlling the influence of the eigenvector similarity on the overall reconstruction effect.

### 3.2 FINDING SHAPE CORRESPONDENCES

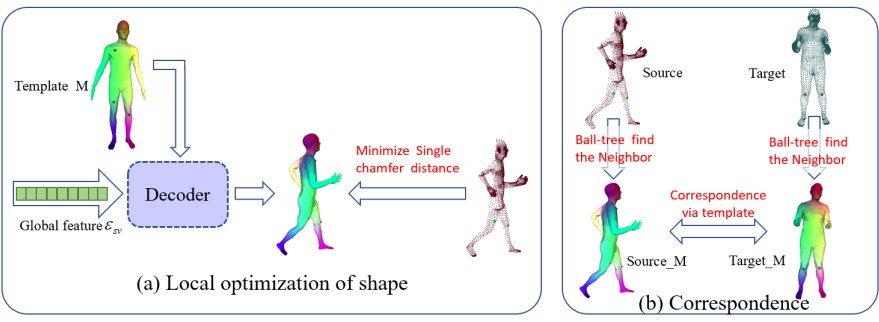

Figure 3: Finding shape correspondences in inference procedure

In the inference procedure, a reconstructed parametric human body model is obtained for each input model by the trained encoder and decoder. This procedure comprises a global encoding and decoding of the human body and can achieve the overall correspondence of the human body; however, the local and detailed parts of the human body are only approximated, and it can be improved. The symmetric chamfer distance between the reconstructed model and input model was solved for in (Groueix et al. (2018))to represent the similarity between the input model and reconstructed parametric model; and the global features of the encoder network output were optimized over 3000 iterations to minimize this chamfer distance. Because our model needs to be applicable to both the single-view human model and complete human model, we use the one-sided chamfer distance to achieve the local optimization between the reconstructed parametric human model and input model and further improve the accuracy of the reconstruction (shown in Fig. 3(a)). The same process can be used to obtain the target human model after the local optimization, and the one-sided chamfer distance loss is as follows:

$$L^{CD}\left(\varepsilon_{sv}\right) = \sum_{p \in M} \min_{q \in S_{sv}} |D_\theta\left(p; \varepsilon_{sv}\right) - q|^2 \tag{5}$$

To obtain the correspondence between the two 3D input shapes (source and target), we first obtain more accurate parametric deformation human models (source M and target M) through the shape deformation network and local shape optimization, and then use the known correspondence between the parametric human models to find the correspondence points. The source and Source M are models of the same pose and body type, and we use a ball-tree to find the nearest neighbors as the corresponding points. The source M and target M are different poses and body types, but they are both parametric models derived from the template deformation, and the correspondence is known. Similarly, the target M and target are human bodies with the same pose and body type, and the ball-tree is used to find the nearest neighbor as the corresponding point. Thus, the correspondence between the source and target can be accurately identified.

## 4 EXPERIMENT

### 4.1 EXPERIMENTAL SETUP

**Training data** Unlike many current human reconstruction methods that rely on a large number of scanned 3D mannequins as training data, our method trains directly on synthetic SMPL data. The SMPL model is a state-of-the art generative model for synthetic humans. The SMPL data can be adjusted to generate a large amount of human data with different poses and sizes. Moreover, the corresponding relationships of parameterized data are known, which reduces the workload for manual labeling. First, we manually deleted a part of the point cloud behind the SMPL standard model under a T-pose, and recorded the index of the deleted point cloud. Then, we extracted the point cloud data of "230 K SMPL" with a large variety of realistic poses and body shape models generated by 3D-coding as the training ground truth. Finally, parts of the back point cloud of all models was deleted according to the recorded index to simulate the generation of the single-perspective point cloud data as the input to training. The T-Pose SMPL standard model and part of the training input with the back point removed are shown in Fig. 4.

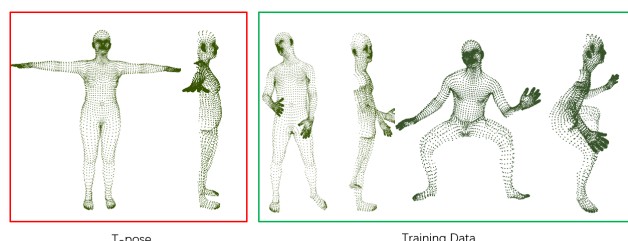

Figure 4: Single perspective point cloud and part of training data

**Test data** We conducted experiments on four human datasets completely different from the training data: FAUST(Bogo et al. (2014)), SCAPE(Anguelov et al. (2005)), SMPL(Loper et al. (2015)),

SMPL-expressive (SMPL-X) (Pavlakos et al. (2019)), and SHREC'19 (de Benoist).

**evaluation criterion** For the evaluation of the corresponding relationships between parameterized human body models of the same type, we calculated the Euclidean distance between the corresponding point pairs of the network outputs as our evaluation index for SHREC'19. We calculated the distance between the target model points in the corresponding point pairs of the network output and real points of the dataset as our evaluation metric, as shown in Equation . Below, $x_i$ is the serial number of the corresponding point of the target human model, $f(x_i)$ is the three-dimensional coordinate of the corresponding point of the target human model, and $gt$ is the three-dimensional coordinate of the ground truth.

$$err = \frac{1}{n} \sum_{x_i \in T} \|f(x_i) - gt_i\|_2 \tag{6}$$

## 4.2 EXPERIMENTAL RESULT

### 4.2.1 SINGLE-VIEW 3D HUMAN BODY CORRESPONDENCE

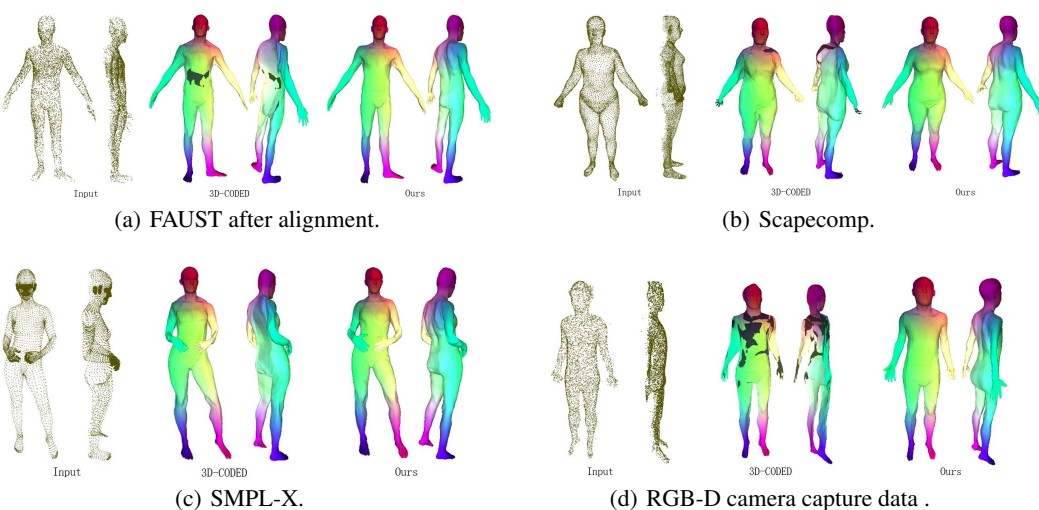

(a) FAUST after alignment.

(b) Scapecomp.

(c) SMPL-X.

(d) RGB-D camera capture data .

Figure 5: Qualitative comparison for the single-view 3D human correspondence task on defferent datasets.

To qualitatively and quantitatively evaluate the reconstruction and corresponding effect of our proposed feature constraint deformation network method on the single-view point cloud input, we use different parameterized human data sets to generate single-view point cloud data for evaluation. The specific data used included the registered FAUST dataset, registered Scapecomp dataset, and SMPL-X model. The corresponding single-view point cloud for testing was generated in a similar way as the training data. To increase the randomness of the data, each dataset was manually generated with its own deletion index to simulate the uncertainty of the edges when the RGB-D camera acquires the single-view point cloud. In addition, we also used a set of RealSense camera data collected from real scene single-view human point cloud data to test the effect. The qualitative and quantitative comparisons are shown in Fig. 5 and Table 1, respectively. As there is no corresponding parametric model for the Realsense data, we only show the results of the qualitative analysis. For the quantitative comparisons, we can compare the corresponding errors of the frontal (visible) part to better reflect the performance of the different methods.

The qualitative comparison is shown in Fig. 5. In Fig. 5(a), it is demonstrated that the 3D-CODED approach reconstructs the human body with a depressed back and the normal of the back part of the reconstructed human model is opposite to the normal of the surrounding vertices; whereas, our method is able to reconstruct the complete human body without these problems as shown in Fig. 5(b). The hand and arm parts of the 3D-CODED human model are severely deformed and the problem of the opposite vertex normal remains. Our method reconstructs the complete human body without deformation because there is no data for a clenched fist in the training data, and our method

Figure 6: Qualitative comparison on scans of Faust.

Figure 7: Qualitative comparison on scans of SHREC'19

reconstructs the human hand as open. From Fig. 5(c), it is clear that our network outperforms 3D-CODED for the SMPL-X human model data, especially for the hands, back, arms, and legs. In the real application scene in Fig. 5(d), the reconstruction effect of the single-perspective point cloud mannequin as photographed by Realsense is more evident. This indicated that our network can reconstruct the single-perspective point cloud mannequin into a more detailed and complete mannequin, and there is no reverse normal situation, whereas the mannequin reconstructed by 3D-CODED produces serious deformations in the arms, hands, and face, and the normal of the body parts the opposite problem is serious. Thus, our method should be more applicable to real scenes.

For the quantitative comparison, we calculated the average Euclidean distance between the self-contained correspondence and computed correspondence as the error. As seen from the experimental results in Table 1, for the reconstruction and correspondence of the visible part, our method outperforms the 3D-CODED on all four datasets.

Table 1: Visible Part Euclidean mean error (EME)

| Dataset | Method | EME |
|---------|--------|-----|
| FAUST | 3D-CODED | 0.371003 |
| FAUST | Ours | 0.362244 |
| SCAPE | 3D-CODED | 0.369555 |
| SCAPE | Ours | 0.367941 |
| SMPL | 3D-CODED | 0.037367 |
| SMPL | Ours | 0.031093 |
| SMPL-X | 3D-CODED | 0.414028 |
| SMPL-X | Ours | 0.413197 |

### 4.2.2 CORRESPONDENCE OF SCANNED 3D HUMAN

**Qualitative testing**

In this subsection, we describe qualitative tests concerning the reconstruction and correspondence of the proposed method on all types of scanned human data, and the results are shown in Figs.6-7.

In Fig. 6, the input comes from the test set in the FAUST scanning data. As shown in Fig. 6(a), the target human hand is placed behind the head, and 3D-CODED only relies on reconstruction of the loss to train the deformation network. As a result,the hand and head are modeled as a whole. As shown in Fig. 6(b), there are some missing parts of the target human feet. Both our method and 3D-CODED can recover the missing parts of the feet; for the hands integrated with the body, our method can also achieve relatively accurate modeling and correspondence.

Fig. 7(a) and Fig. 7(b) are derived from the results of the Princeton dataset (Chen et al. (2009)) in the SHREC'19 dataset (de Benoist). As can be seen from the results for Fig. 7(a), which presents the human body with a more ambiguous input, our proposed method can serve to complement the detailed parts of the human face and can reconstruct a more reasonable human body than 3D-CODED. As shown in Fig. 7(b), our method can reconstruct reasonable hand and face information, whereas 3D-CODED reconstructs the human body with three fingers, facial layering, and other errors.

**Correspondence error analysis**

To further validate the performance of the proposed method in finding correspondences, we performed a quantitative comparison with the existing state-of-the-art methods on the FAUST and SHREC'19 datasets, as these provided challenges and true values of the data.

In the FAUST dataset-INTER challenge, the test data comprised scanned complete human models with different postures, and the comparison method used a complete human dataset as the training

Table 2: FAUST dataset INTER challenge performance of different methods

| Serial number | Method | Error(cm) |
|---|---|---|
| 1 | Convex-Opt (Chen & Koltun (2015)) | 8.304 |
| 2 | FMNet (Litany et al. (2017)) | 4.826 |
| 3 | FARM (Marin et al. (2020a)) | 4.123 |
| 4 | LBS-AE (Li et al. (2019)) | 4.079 |
| 5 | Smooth Shells (Eisenberger et al. (2020)) | 3.929 |
| 6 | BPS (Prokudin et al. (2019)) | 4.529 |
| 7 | SP_inter_challenge (Zuffi & Black (2015)) | 3.126 |
| 8 | Adversarial GP network (Zhou et al. (2020)) | 2.759 |
| 9 | 3D-CODED (Groueix et al. (2018)) | 2.878 |
| 10 | LoopReg (Bhatnagar et al. (2020)) | 2.660 |
| 11 | Learning element structure (Deprelle et al. (2019)) | 2.578 |
| **12** | **Ours (single-view shape as input)** | **2.774** |
| **13** | **Ours (complete shape as input)** | **2.541** |

data. To be fair, we trained two versions of the model: one version was the same as in all of the previous experiments and its input data was the synthetic point cloud data that simulated the single view and deleted the backside points; and the input data of the other version used the full SMPL parameterized model as the ground truth. The experimental results are shown in Table 2. It can be seen that the training results from our method on the synthetic single-perspective point cloud are better than most comparison methods (including 3D-CODED). When the input data is a complete human body during training, our method obtains the lowest error, outperforming the other baseline methods.

Similarly, on the SHREC'19 dataset (de Benoist), we tested both versions of the model and compared the results with those given on the challenge website, as shown in Table 3. Because this dataset contains 3D human models with varying densities from 5,000 to 50,000 vertices, and with uniform and non-uniform distributions, the models are highly variable; therefore, the overall test results are worse than those for the FAUST dataset. However, our proposed method, trained on the input as both a synthetic single-view point cloud and complete human point cloud, outperforms the comparison method, demonstrating the good generalization capability of the proposed method in terms of finding correspondences between different human bodies.

Table 3: Performance on SHREC'19 dataset

| Serial number | Method | EME(cm) |
|---|---|---|
| 1 | 3D-CODED (Groueix et al. (2018)) | 8.1 |
| 2 | Diff-FMaps (Marin et al. (2020b)) | 7.1 |
| 3 | Elementery Structures (Deprelle et al. (2019)) | 7.6 |
| 4 | CorrNet3D (Zeng et al. (2021)) | 6.9 |
| **5** | **Ours (single-view shape as input)** | **6.83** |
| **6** | **Ours (complete shape as input)** | **6.62** |

## 5 CONCLUSION

We propose a network structure that can be applied not only to find correspondences between complete human models, but also to find correspondences between complete human models and single-view 3D human models. The key point of our method is the use of feature vector similarity and single-side chamfer distance to further constrain the reconstruction; this will improve the reconstruction quality and correspondence search accuracy. The proposed method improves the accuracy in finding the correspondence between the complete human body model and single-view human body model. It achieves errors of 2.774 cm and 2.541 cm on the single-view and complete shapes as inputs on the FAUST dataset, respectively, and of 6.83 cm and 6.62 cm on the SHREC'19 dataset, respectively.

ACKNOWLEDGMENTS

This work was supported by National Natural Science Foundation of China (No. 61901436)

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
