# OpenReview forum: "Deep Deformation Based on Feature-Constraint  for 3D Human Mesh Correspondence"
_ICLR.cc/2023/Conference — Submitted to ICLR 2023_

### Official Review · Reviewer_KeGj · 2022-10-23

**Confidence:** 3
**Correctness:** 4
**Technical Novelty And Significance:** 2
**Empirical Novelty And Significance:** 2
**Recommendation:** 5

**Clarity, Quality, Novelty And Reproducibility:**

As it was commented above, the paper is easy to follow, the novelty is a bit limited due to the work being incremental with the literature and the main contributions and differences with some works are not provided in the paper; and the experimental results are enough. Considering the main paper, reproducibility is a bit hard as some details and parameters are missing.

**Strength And Weaknesses:**

In general, the paper is well written and it is clear enough. The motivation is clearly presented, and the discussion of competing approaches seems to be correct.

q_j needs to be introduced in Eq. (1).

Figure 1 should include a self-contained caption. To be honest, every figure in the paper should consider that.

According to the learning and inference pipelines, I think the proposed method is a bit incremental. The idea of deforming a template by considering 2D observations is a well-known problem. Once the template is deformed, the initial human shape is fully estimated. To produce this type of results, the authors assume in training a template at rest and a complete shape in a particular state. In my opinion, the amount of 3D priors to learn the model is very high. At the end, this could be reduced to an alignment task with different sets. However, as the visible points in the set can be observed in the input image, this problem is tractable. In particular, the authors should clarify the differences with Groueix et al. (2018).

Similar results in terms of a chamfer distance could be derived from very different 3D poses, due to the projection ambiguity on deformable bodies. How can that be handled?

The method provides competitive solutions with respect to a wide variety of competing approaches. Both quantitative and qualitative evaluation are well executed. I miss some experiments with large occlusions in the input data, the presence of holes, and so on. That means I would like to see more challenging and realistic scenarios.

What about the scale? Could very different bodies be matched? For example, using the current template, incorporating a child as an input data.

**Summary Of The Paper:**

In this paper the authors propose a neural model to deform a template based on the single-view 3D observations while correspondences between both states are implicitly computed. To that end, the authors exploit a feature constraint deformation network on the single-view point cloud as input. The method uses well-known ideas and modules in literature, providing competitive results with respect to an ample variety of approaches.

**Summary Of The Review:**

On balance, the method is simple yet effective, the results seem to be promising, but the technical contribution is a bit incremental. As pointed out, Some relevant experiments and details could improve the paper.

---

### Official Review · Reviewer_tJR5 · 2022-10-24

**Confidence:** 5
**Correctness:** 2
**Technical Novelty And Significance:** 2
**Empirical Novelty And Significance:** 2
**Recommendation:** 3

**Clarity, Quality, Novelty And Reproducibility:**

The paper is reasonably well written and I could follow it easily. I have some concerns regarding originality (see weakness section). It will be very useful if authors clarify why is their method different than just putting cosine similarity (between features of partial and complete shape) on existing 3dcoded network.

**Details Of Ethics Concerns:**

I do not see immediate ethical concerns arising from the work.

**Strength And Weaknesses:**

Strengths:
+ The task of registering partial/ single view data with a common template is quite challenging and interesting.
+ The proposed methodology is simple and makes intuitive sense.

Weaknesses:
- I find the experimental setup (Sec 4.1) problematic. The paper claims to work with single view point clouds, but this is actually not the case. Authors use a very wrong way to generate single view data. They remove the back of the SMPL model in canonical pose and repose the remaining vertices to get the training data. This is not true for actual single view data. *The parts that are missing in actual single view data are the ones that are occluded from the camera view due to occlusion not a fixed set of vertices*. The most common source of occlusion are hands as they have high degree of freedom and can cover various parts of body. In this work the hands and hand-related occlusions are totally ignored.
Model trained with this data would not have to reason about the occlusion at all and can just memorise how to complete a smooth back, which is fixed as fixed SMPL vertices are always removed.
This process is used both for train and testing, making the evaluation highly unreliable.

- Technical  novelty appears to be limited. The encoder-decoder formulation and template deformation look very similar to 3D-CODED. The new component is enforcing that the features from the partial point cloud and the complete point cloud should be same. This is enforced via cosine similarity on the encoded features. Can the authors comment on this?

- Authors primarily compare their method with works that deal with full point cloud registration. Works like IPNet, ECCV'20 (not cited) are more suitable baselines as they have shown that they can complete the missing point cloud using implicit functions and then perform 3D registration.

- Why not report results on FAUST Intra?

Clarifications:
- The feature decoding and per-vertex deformation approach is very similar to 3DCODED, which requires several initialisations to handle global orientation. Is this also the case for the proposed approach?

- Do the authors use the official FAUST test set (GT not available, but results need to be submitted to their portal for evaluation)? I've seen a few works overfitting to FAUST train set and reporting results. This is a bit of cheating :P Just want to make sure.

- More recent works like LoopReg, NeurIPS'20 and IPNet, ECCV'20 show registration on dressed human shapes, which is more challenging than the current setup where synthetic SMPL is used. Can the authors comment why are they restricting themselves to this old setting?

Minor suggestions:
- Please make figure captions more informative.
- Fig. 2 just shows cosine similarity loss. Is it really necessary?
- Typo: "The source and Source M are" below eq. 5

**Summary Of The Paper:**

The authors address the task of 3D human mesh registration with key focus on registering a complete template to (incomplete) single view point cloud. Authors show better performance than baselines on the registration as well as correspondence prediction task.

**Summary Of The Review:**

I have serious concerns with the way authors generate single view data. This is not how single view data is generated and this is the primary contribution of the work (handling single view data). The novelty also appears to be limited (see weakness) as compared to 3dcoded but I'd like to hear from authors on this.

---

### Official Review · Reviewer_waCJ · 2022-10-24

**Confidence:** 4
**Correctness:** 2
**Technical Novelty And Significance:** 3
**Empirical Novelty And Significance:** 3
**Recommendation:** 3

**Clarity, Quality, Novelty And Reproducibility:**

The main idea is quite simple yet effective as the authors back by the quantitative experiments. The method is simple enough to be reproducible, but it would help if the authors provided all the implementation details (including the choice of optimization strategy, learning rate, choice of the constant b etc.).

**Strength And Weaknesses:**

# Strengths
The method shows simple yet effective approach to extend a template based human body reconstruction/correspondence estimation method to work on partial observations and report quantitative SotA results.

# Weaknesses
## Related Work
The main focus of the method is to support shape correspondence estimation from single-view (RGBD) observations. While the Related Work section discusses various methods for human shape correspondence estimation, it never makes a distinction between the methods which require a full scan on the input and the ones which can handle single-view input only. The authors should position this paper within the SotA so that it is clear what work was already done and who are the competitors. The authors only mention: "However, these methods are usually proposed for matching complete 3D human models". What do the authors mean by "usually"? Do some of the  methods in fact support a single-view input? If so, which ones are those? The authors further mention: "there are relatively few methods applicable to single-view human body data collected by a single RGB-D camera." However, the authors never discuss any such method.

## Methodology
- Fig. 2: The Figure seems redundant as the notion of cosine similarity is well known. I would suggest removing the Figure in favor of more substantial content (e.g. experimental results).
- The last paragraph - the explanation of correspondence retrieval: The text is quite confusing and it is not clear how exactly the authors find the correspondences. The arrows in Fig. 2 indicate that the points are projected from the template to the observations, but more typical approach would be to project a point from observation A to the template and then from the deformed template to the deformed observation B. This would be clarified if the authors perhaps formalized the process with a mathematical formula.

## Experiments
- "Unlike many current human reconstruction methods that rely on a large number of scanned 3D mannequins as training data" -> Please provide citations for this claim.
- The paragraph "evaluation criterion": Please formulate and clarify the paragraph as it is unclear what your metric is and whether it applies in the same way for all the datasets.
- Fig. 5: The caption talks about a 3D human correspondence task, but the image shows a reconstruction instead.
- Figures 5, 6, 7 - If a reconstruction of only a single frame is shown, the color coding of the human does not show any useful information unless it is directly compared to the GT color coded human shape.
- The authors never show qualitative comparison of correspondences between various body poses despite this application being advertised in the Abstract.

## Presentation
In general, the paper is quite difficult to read, due to the obscure formulations the authors choose to use and it is thus often unclear what the authors had in mind. Some examples are listed below but the paper contains many more of these:
Statements where it is hard to understand what the authors mean:
- "The parametric human model has been widely used in various human-related applications and in 3D human mesh correspondence because it provides sufficient scope to modify the resulting model." - What do the authors mean by "sufficient scope to modify the resulting model"?
- "In representation learning, one of the main challenges is to design appropriate loss functions for supervising features with different abilities" - What do the authors mean by "supervising features with different abilities"?
- "Meanwhile, the decoder generates a completely transformed template with higher promise" - What do the authors mean by "higher promise"?
- "The deformed models have the same vertex definition, definite semantic information, and same face connections as the template." - What do the authors mean by "same vertex definition"?
- "Motivated by Deng et al. (2019), we attempted to penalize the loss in the feature space," - Perhaps better formulation would be "penalize the XYZ by adding a loss ABC".
- "this feature is taken as the observed quantity in Equation (1) and in the human template as an input pair to output the deformed human shape vertices." - What input pair?
- " then constrain the single-view human shape and intact human shape in the feature space as close as possible, so as to use the same network framework for both the single-view and full-body shapes. " - Please reformulate (the use of "so as to").
- "Here, b is a scale coefficient for controlling the influence of the eigenvector similarity on the overall reconstruction effect." - The use of any eigenvectors was not either mentioned until this point or after. What eigenvectors do the authors have in mind?
- "Below, xi is the serial number " -> Perhaps index number?

Furthermore, there are plenty of unjustified or unexplained statements which come across as filler sentences rather than content of substance, unless reformulated and/or better cited. Some examples are listed below:
Unjustified, unexplained or incorrect statements such as:
- "The obtained network can better realize the deformation and reconstruction of the single-view human body." - Better in what sense?
- "The proposed method selects a pa- parameterized human body model with a standard pose as a template," - What is the "standard pose"?
- "deforms it into source and target models with arbitrary poses and body shapes, respectively." - It was never explained what are the source and target models.
- " which reduces the workload for man- ual labeling. " - It does not reduce the workload but removes it completely.

## Miscellaneous:
- The authors use the term "input model" to refer to the input data/observations, which might be confusing to the reader. Please reformulate.
pg. 6: "source M and target M" - Calling the source and target as source M and target M sounds confusing, perhaps use e.g. M_{s}, M_{t}.

## Questions for the authors:
Eq. 4: Why did the authors choose to formulate the final loss as a convex combination of the two members Lr and Lf, rather than a more common way to combine the individual loss terms involving a linear combination, e.g. Loss = LR + b * LS?

Typos:
- pg1: "This makes the correspondence process more challenging." -> "This makes the correspondence estimation process more challenging."
- pg2: "global feature coded from the input shape." -> "global feature encoded from the input shape."
- pg.4: "where,pj denotes" -> "where pj denotes"
- pg.4: " simplified PointNet network" -> addd citation
- pg.5: "where,M" -> "where M"
- pg.5: "whereSc  -> "where Sc"
- pg.5: "it is will be unable " -> "it will be unable "
- pg.6: "(Groueix et al. (2018))to" -> "(Groueix et al. (2018)) to"
- pg.6: "EXPERIMENT" -> "EXPERIMENTS"
- pg.7: "shown in Equation . " -> "shown in Equation 6"


**Summary Of The Paper:**

The authors present a method to estimate human body shape correspondence from both the full scans and from single-view depth data. The authors build on a well established method 3D-CODED but they add a mechanism to support partial depth observations as well. Specifically, the authors train an encoder extracting a global code both on the full scans and on corresponding partial observations and introduce an additional loss function to force the global code to remain the same.

**Summary Of The Review:**

While the authors present a simple and effective method which yields SotA results on the chosen datasets, the paper presentation is of rather low quality. Specifically, the authors often use confusing and unclear formulations, filler statements and unjustified claims. There are quite a few typos (some of them listed above). Some parts are only explained in words in a bit confusing way where a simple mathematical formulation would help (see Weaknesses). The figures presented in the Experiment section use a color coding which does not visually help unless a GT is provided next to them as well. Overall, I believe the quality of the submission is not up to the standard of the ICLR and the paper could use some more polishing.

---

### Official Review · Reviewer_PTZp · 2022-10-25

**Confidence:** 4
**Correctness:** 2
**Technical Novelty And Significance:** 2
**Empirical Novelty And Significance:** 1
**Recommendation:** 3

**Clarity, Quality, Novelty And Reproducibility:**

The FCDnet do not have clear Novelty. The overall framework is simple combination.

**Details Of Ethics Concerns:**

The paper use the mature dataset to train and test. It should not have ethic problems.

**Strength And Weaknesses:**

Some major problems are show below.

1. The main contribution of the paper is unclear. The author did not present it explicitly. Can not find it anywhere.

2. The author did a severe mislead on quantitative analysis on "MPI Faust dataset". For most of the experiments, the author did the comparison with 3D-CODED, which is an ECCV 18 paper. I have checked the MPI FAUST dataset for Inter-subject challenge, the SOTA can achieve 1.164cm, at least 5 methods (2 of them are already published, 1 has released the code[1]) has better results than current paper.

3. In the test data, what's the SMPL and SMPL-X mean? From my point of view, they are not dataset, they are just parametric model. Based on SMPL or SMPL-X, the community create several datasets, such as SURREAL or AMASS.

4. The FCDnet do not have clear Novelty. Either end-to-end framework or feature constraints have been widely used.

5. The figures are not clear, please emphasize the difference.

[1] Deep Virtual Markers for Articulated 3D Shapes. ICCV 2021
[2] Learning from Synthetic Humans (SURREAL). CVPR 2017
[3] AMASS: Archive of Motion Capture As Surface Shapes. CVPR 2019

**Summary Of The Paper:**

The paper propose a feature constraint deformation network (FCDNet), the networked has tested on different benchmarks.  No clear contribution for this work.

**Summary Of The Review:**

Due to the overall major problems, the paper needs to major revision and make the contribution more clear and offer the fair comparison before publication.

---

### Decision · Program_Chairs · 2023-01-20

**Decision:**

Reject

**Justification For Why Not Higher Score:**

weakness, no author response

**Justification For Why Not Lower Score:**

n/a

**Metareview: Summary, Strengths And Weaknesses:**

Paper proposes a method to register 3D meshes of the human body to (single-view) depth maps by building on 3D-CODED.  Main contribution is to handle partial depth observations.

Strengths:
+ simple and effective, leads to good results

weaknesses:
- weak and incremental technical contribution
- clarity: formulation unclear and confusing, unjustified statements


**Summary Of Ac-Reviewer Meeting:**

n/a